# Photobiomodulation Therapy on Brain: Pioneering an Innovative Approach to Revolutionize Cognitive Dynamics

**DOI:** 10.3390/cells13110966

**Published:** 2024-06-03

**Authors:** Tahsin Nairuz, Jong-Ha Lee

**Affiliations:** Department of Biomedical Engineering, Keimyung University, Daegu 42601, Republic of Korea; tahsin.bmb@nstu.edu.bd (T.N.); sangwoo.cho83@gmail.com (S.-C.)

**Keywords:** photobiomodulation therapy, cytochrome c oxidase, Parkinson’s disease, stroke, cognitive functions, transcranial approach, intracranial and intranasal delivery

## Abstract

Photobiomodulation (PBM) therapy on the brain employs red to near-infrared (NIR) light to treat various neurological and psychological disorders. The mechanism involves the activation of cytochrome c oxidase in the mitochondrial respiratory chain, thereby enhancing ATP synthesis. Additionally, light absorption by ion channels triggers the release of calcium ions, instigating the activation of transcription factors and subsequent gene expression. This cascade of events not only augments neuronal metabolic capacity but also orchestrates anti-oxidant, anti-inflammatory, and anti-apoptotic responses, fostering neurogenesis and synaptogenesis. It shows promise for treating conditions like dementia, stroke, brain trauma, Parkinson’s disease, and depression, even enhancing cognitive functions in healthy individuals and eliciting growing interest within the medical community. However, delivering sufficient light to the brain through transcranial approaches poses a significant challenge due to its limited penetration into tissue, prompting an exploration of alternative delivery methods such as intracranial and intranasal approaches. This comprehensive review aims to explore the mechanisms through which PBM exerts its effects on the brain and provide a summary of notable preclinical investigations and clinical trials conducted on various brain disorders, highlighting PBM’s potential as a therapeutic modality capable of effectively impeding disease progression within the organism—a task often elusive with conventional pharmacological interventions.

## 1. Introduction

Photobiomodulation (PBM) constitutes a sophisticated approach rooted in utilizing visible and near-infrared light at minimal power densities, and represents a captivating frontier in medical treatment. This application necessitates precise control over the administered fluence to ensure the avoidance of thermal or ablative consequences. PBM operates on an intricate photochemical mechanism, engaging with specific chromophores residing within cellular constituents. Notably, cytochrome c oxidase (CCO), an integral player in the electron transport chain (complex IV) of mitochondria, exhibits absorbance within the wavelength range of approximately 600 to 900 nm. Beyond this range, longer wavelengths are theorized to engage with water molecules and light-gated ion channels, especially belonging to the transient receptor potential (TRP) family, thereby inducing their activation. Moreover, PBM demonstrates the remarkable capability to mitigate oxidative stress through dislodging inhibitory nitric oxide (NO) from the CCO complex, thus orchestrating a cascade of intricate cellular responses conducive to therapeutic outcomes [1].

In 1967, PBM was serendipitously stumbled upon by Endre Mester of Hungary, although the principles of light therapy had been acknowledged earlier. His pursuit was to replicate an experiment conducted by McGuff from the USA [2], who utilized a ruby laser beam to target a cancerous tumor in a lab rat [3]. Despite Mester’s ruby laser being significantly less powerful than the one used by McGuff, it yielded unforeseen results. Instead of eradicating tumors, Mester observed remarkable outcomes: a stimulation of hair regrowth and accelerated wound healing at the tumor sites [4,5]. This serendipitous finding marked the inception of what Mester termed “laser biostimulation”, subsequently recognized as “low-level laser therapy” (LLLT) [6,7].

Throughout the past twenty years, brain PBM therapy has become a pioneering approach aimed at augmenting neural activity to enhance cognitive function. This innovative technique involves exposing neuronal tissue to light, typically from less than 1 to over 20 J/cm^2^, and spanning wavelengths from the red spectrum to the near-infrared (NIR) region, between 600 and 1100 nm [8]. The delivery of this light is facilitated through diverse methodologies tailored to ensure the precise targeting of neural substrates. PBM operates intricate chemical cascades within neuronal cells, thereby orchestrating a spectrum of biological responses that encompass a gamut of physiological enhancements, ranging from the fortification of neuroprotective mechanisms and metabolic optimization to the facilitation of neurogenesis and the attenuation of inflammatory processes and oxidative stress [9].

In the current state, transcranial PBM (applying light directly to the head for treating neurological ailments) research remains in an incipient phase, characterized by a lack of extensive clinical validation despite the accumulation of compelling evidence from animal studies. Nevertheless, the available literature heralds a promising outlook, suggesting that PBM holds substantial promise as a therapeutic avenue for a diverse array of neurological disorders, spanning from stroke and traumatic brain injury (TBI) to Parkinson’s disease (PD), Alzheimer’s disease (AD), and depression, as well as potentially offering cognitive augmentation among healthy individuals [10]. The intricate molecular mechanisms driving the effects of PBM remain largely elusive. This review therefore endeavors to shed light on the mechanistic, neurobiological, and biophysical underpinnings of the brain PBM response across various preclinical and clinical investigations.

## 2. Methodology

To critically evaluate current advancements in the biological, photophysical, and clinical dimensions of PBM, we conducted a narrative literature review concerning the application of PBM therapy on the brain, emphasizing its potential to revolutionize cognitive dynamics. We meticulously selected clinical studies centered on brain PBM therapy suggesting significant improvements in memory, executive function, and mood disorders, as well as promising therapeutic avenues for stroke, TBI, and neurodegenerative diseases. Moreover, to comprehensively elucidate the photobiological aspects and the effects of PBM on brain function, we considered both original research articles and review papers. The selection process encompassed searches across multiple library databases, including PubMed, Scopus, Web of Science, and Google Scholar, focusing on publications from 2010 to February 2024. The search was conducted using medical subject heading terms and free-text terms. The search strategy employed the following descriptors and Boolean operators: “Photobiomodulation OR PBM OR LLLT OR Low level Laser Therapy” AND “brain OR neurotherapy” AND “cognitive function OR neuroprotection” AND “clinical trial OR clinical study”; “Photobiomodulation” AND neuron; “Photobiomodulation” AND neurodegenerative disease. Additionally, we included the references of reviewed publications to ensure a comprehensive inclusion of relevant studies. The preliminary screening phase entailed a review of titles and abstracts to eliminate irrelevant and duplicate studies. Studies that did not address the proposed theme, those lacking full text availability, or those published in potential predatory journals were excluded from this review.

## 3. Molecular Mechanisms Underlying PBM Therapy

### 3.1. Mitochondrial Cytochrome C Oxidase

The predominant mechanism underlying PBM revolves around cytochrome c oxidase (CCO), an essential component in the respiratory chain of mitochondria (Figure 1). CCO, composed of two copper and two heme centers with different spectra for light absorption, serves as the final enzyme of complex IV of the mitochondrial electron transport chain. It helps release electrons from cytochrome c towards molecular oxygen, which converts it into water through glucose metabolism. In circumstances marked by hypoxia or cellular damage, nitric oxide (NO) can impede CCO enzyme activity. However, photons absorbed by CCO, especially within the red (600–700 nm) and near-infrared (760–940 nm) spectral ranges, can dissociate inhibitory NO [9]. This dissociation event triggers a cascade of effects: a surge in mitochondrial membrane potential, enhanced oxygen consumption, and intensified glucose metabolism, culminating in an amplification of ATP production within the mitochondria.

### 3.2. Light/Heat Sensitive Ion Channels

Even though the existing data backing light- or heat-gated ion channels as the processes driving PBM remain somewhat limited, ongoing research is steadily expanding in this area [11]. There is speculation that PBM could impact transient receptor potential (TRP) channels, which are calcium channels sensitive to various stimuli, including light and heat [12].

Among TRP channels, the TRPV “vanilloid” subfamily has received significant attention in PBM research. Recent studies suggest that green or infrared light may induce the activation of TRP channels [13,14,15] (Figure 1). Nonetheless, while green light has less practical clinical use due to its limited tissue penetration, infrared and near-infrared light are more promising. Research indicates that exposure to specific infrared wavelengths, such as 2780 nm, can attenuate TRPV1 activation, reducing pain stimulus generation. Similarly, TRPV4 has been found to be responsive to pulsed light at 1875 nm [14]. As water predominantly absorbs infrared light in this spectral range and is abundantly present as the primary molecule in living organisms, even subtle changes in temperature, below the threshold for tissue heating detection, could conceivably induce alterations in protein conformation, potentially leading to the opening of ion channels.

### 3.3. Retrograde Mitochondrial Signaling

PBM is thought to elicit retrograde mitochondrial signaling, an intriguing process where communication occurs between mitochondria and the cell nucleus, contrary to the typical flow of signals from the nucleus to other organelles [16]. This bidirectional communication is facilitated by alterations in the mitochondrial ultrastructure that trigger a cascade of intracellular changes, affecting ATP synthesis, pH, cAMP levels, and intracellular redox potential. Consequently, alterations in membrane permeability and ion flow across the cell membrane occur, triggering the generation of ROS and influencing the function of redox-sensitive transcription factors like activator protein-1 (AP-1) and NF-κB [17] (Figure 1). Moreover, Karu’s insights suggest supplementary pathways, including directly increasing the expression of particular genes, influencing the multifaceted effects of PBM on cellular function and signaling [18].

## 4. Current Strategies for Delivering Light to Different Brain Regions

Transcranial photobiomodulation (tPBM) entails the non-invasive process of delivering light emitted by an external source, like a laser or LED, through several layers of the head, starting from the scalp, periosteum, skull bone, meninges, and dura, ultimately reaching the surface of the brain cortex [19] (Figure 2A). However, due to the rapid decrease in the intensity of light as it traverses through these tissues, only a fraction of the incident light effectively reaches the cortex’s outermost layers. This limited penetration poses a challenge to providing deeper brain regions with an adequate dose of light [20]. To address this challenge, combined treatments involving nanoparticles and light [21,22] and the near-infrared time-reversed ultrasonically encoded (TRUE) technique [23] aim to maximize light penetration into the brain. Additionally, a recent approach utilizing transcranial multi-directional irradiation with multiple LED arrays has been shown to enhance the photon flux of the cerebrum and achieve a more uniform distribution of photons throughout the brain, all without a significant increase in temperature [24]. In addition, using a high-power laser (Class 4, with power ranging from 10 to 15 W) rather than a low-power laser (Class 3, typically less than 0.5 W) has been proposed as a means to provide a larger fluence of light to deeper cortical layers instead of causing thermal harm [25,26].

The dysfunction of neurons and circuits within the globus pallidus internus (GPi) and subthalamic nucleus (STN), two deeper parts of the brain, contributes to Parkinson’s disease (PD) pathogenesis [27]. However, the limited penetration of light into these deeper brain areas presents a barrier to delivering an effective light dose to midbrain neurons. The latest attempts have focused on developing a technique to deliver light to deeper tissues in the brain, like the substantia nigra pars compacta (SNc), by employing an intracranial method akin to deep brain stimulation [28] (Figure 2B). This technique involves implanting an optical fiber for light delivery, which has been utilized in both PBM [29] and optogenetic studies [30]. Studies by Monte Carlo have demonstrated that light delivery efficiency to the SNc is substantially higher with a fiber optic source inserted in the third ventricle than with non-invasive transcranial methods [31]. Furthermore, preliminary studies examining the neuroprotective impacts of intracranial PBM have not revealed any negative effects in the vicinity of the implant locations within the midbrain [32].

While most researchers have concentrated on transcranial and intracranial light delivery methods [33,34], there is also promising evidence that light radiation via the nasal and oral cavities can improve symptoms of dementia and PD [35,36,37] (Figure 2C,D). In the intranasal PBM approach, light is positioned within the nostril, allowing the direct irradiation of subcortical and cortical structures associated with the pathologies of AD and PD [38]. Additionally, combining light exposure through the sphenoid sinus and mouth cavity has been proposed to achieve sufficient fluence at the SNc in humans [31]. Notably, a recent case series demonstrated cognitive and functional improvements in dementia patients with a combined intranasal and transcranial LED treatment regimen [35]. Moreover, a proposed method involves using intense LED light at 448 nm that is administered into the ear canal in order to target the human brain’s temporal lobes by entering through the skull [39]. This approach has been considered for PBM therapy aimed at treating seasonally affective disorders [40,41] and modulating brain activity [39].

## 5. Depth of Light Penetration and Sources of Light

Comprehending the optical characteristics, such as absorption and scattering coefficients, of various head tissues enables the pinpointing of the optimal wavelength range for maximal light transmission [42]. Functional near-infrared spectroscopy (fNIRS), utilizing light within the 700–900 nm range, has emerged as a brain imaging technique comparable with a revolutionary impact comparable to that of functional magnetic resonance imaging (fMRI) in recent neuroscience advancements [43]. Various studies have explored the light penetration depths across multiple wavelengths via the scalp and skull, shedding light on their ability to reach into the brain. For instance, Haeussinger et al. evaluated the average penetration depth of NIR light throughout the scalp and skull to be approximately 23.6 ± 0.7 mm [44]. Jagdeo et al. utilized human cadaver heads to assess the penetration of 830 nm light, revealing variations across different anatomical regions of the skull (temporal region: 0.9; frontal region: 2.1%; and occipital region: 11.7%) [45]. Similarly, Tedord et al. conducted experiments with human cadaver heads to compare the penetration capabilities of light at varying wavelengths (660 nm, 808 nm, and 940 nm), finding that 808 nm light exhibited the deepest penetration into the brain, reaching depths of 40–50 mm [46].

Meanwhile, Lapchak et al. conducted a comprehensive analysis of light transfer through skulls across various species, revealing substantial differences: mice exhibited 40% transmission, rats exhibited 21%, rabbits exhibited 11.3%, and human skulls exhibited a mere 4.2% [47]. Meanwhile, Pitzschke et al. delved into the intricacies of light penetration into the brain using both transcranial and transsphenoidal delivery methods, determining that the most effective combination was the transsphenoidal delivery of 810 nm light [31]. Moreover, they investigated the impact of tissue storage and processing methods on optical features in rabbit heads [48]. Similarly, Yaroslavsky et al. scrutinized light penetration through various brain tissues such as white brain matter, gray brain matter, cerebellum, brainstem tissues, pons, and the thalamus, pinpointing optimal wavelengths of 1000 to 1100 nm [49]. Furthermore, Henderson and Morries discovered that within the wavelengths of 810 nm or 980 nm, 0.45% and 2.90% of the light entered 3 cm of lamb scalp, skull, and brain tissue ex vivo [25].

For tPBM, a diverse range of light sources have been utilized, including lasers and LEDs. A contentious issue yet to be definitively resolved is whether or not coherent monochromatic lasers outperform non-coherent LEDs, which typically possess a 30 nm band-pass. Commonly utilized wavelengths range from an NIR of 800 to 1100 nm, although red wavelengths are also employed either independently or alongside NIR. The power levels and laser outputs vary significantly from high-powered Class IV lasers at around 10 W to more moderate 1 W lasers [50]. LEDs exhibit a wide range of total power levels depending on factors like array size and the quantity and power of each of the diodes, with some diodes now capable of reaching up to 3 W each. Additionally, there are significant differences in power densities, for example, the Photothera laser [51] and other class IV lasers [25] require active cooling or continuous motion with power densities around 700 mW/cm^2^, while LEDs have power densities between 10 and 30 mW/cm^2^.

## 6. Neurobiological Consequences of Photobiomodulation Therapy

At the pinnacle of brain photobiomodulation therapy, pivotal outcomes emerge: optimizing cerebral metabolic activity, enhancing blood flow and angiogenesis, mitigating apoptosis while bolstering antioxidant levels and diminishing inflammatory responses, promoting neurotrophins secretion, fostering the activation of neuroprogenitor cells, and facilitating synaptogenesis, thereby enabling the formation of novel neuronal connections (Figure 3).

### 6.1. Neuronal Bioenergetics

Mitochondria undergo significant alterations under pathological circumstances, including diminished activity for respiratory chain complexes and attenuated ATP synthesis, alongside excessive ROS production, mitochondrial membrane potential (MMP) disruption, a transition of permeability in the inner mitochondrial membrane, and the subsequent discharge of cytochrome c in the cytosol [52]. Therefore, mitochondrial malfunction plays a critical role in the progression of numerous neurological and psychological disorders [53,54].

Neural tissue is notably abundant in mitochondria, and is thus highly responsive to light interactions, particularly with CCO, a key player in neuronal energy metabolism. Early investigations into cultured rat visual cortical neurons unveiled that LED light (4 J/cm^2^) at 670 nm and 830 nm effectively upregulated CCO activity [55], with 670 nm light reversing downregulation induced by tetrodotoxin [56]. Further research indicated that in naïve rats’ prefrontal cortex (PFC), LED light (10.9 J/cm^2^) irradiation at 633 nm led to a 14% increase in CCO activity [57], whereas in a rat model with rotenone-induced neurotoxicity, this irradiation resulted in a 26% augmentation in the superior colliculus and a remarkable 60% increase (3.6 J/cm^2^) in CCO activity across the whole brain [58]. Recent research demonstrated that transcranial LED therapy at 808 nm elevated CCO activity in mouse models of stress (41 J/cm^2^) [59] and Alzheimer’s disease (3 J/cm^2^) [60]. Moreover, in transgenic mouse models of AD, a remarkable restoration in CCO expression was observed in the neocortex and hippocampus after transcranial LED therapy for 4 weeks at 670 nm [61].

Neural tissues exhibit profound reliance on ATP generated by mitochondria for energy demands. In various mouse models, including amyloid protein precursor (APP) transgenic mice (6 J/cm^2^) [62], Aβ-induced Alzheimer’s disease (AD) mice (3 J/cm^2^) [60], and a major depression (41 J/cm^2^) mouse model [59], cerebral ATP levels were augmented utilizing transcranial PBM therapy with an 808 nm laser. However, LED treatment (670 nm, 4 J/cm^2^) in a single session elevated ATP levels in striatal neurons exposed to 1-methyl-4-phenylpyridinium (MPP+), while laser irradiation (670 nm, 1 J/cm^2^) in a single session failed to elevate ATP levels in Aβ-treated PC12 cells [63] or in Parkinson’s disease (PD) cybrid cell lines (810 nm, 2 J/cm^2^) [64]. These variations may potentially be attributed to differences in the applied light fluences. Direct exposure of the parietal cortex to laser light (830 nm) also enhanced the ATP/ADP ratio in healthy rats [65]. Notably, research in embolic stroke models of rabbit unveiled that transcranial laser treatment in a single session at 808 nm significantly boosted cortical ATP content, with efficacy observed in both CW mode (0.9 J/cm^2^) [66] and PW mode at frequencies of 100 Hz (4.5 and 31.5 J/cm^2^) [67]. Moreover, in a mouse model of TBI, 10 Hz PW laser light at 810 nm showed efficacy in enhancing cerebral ATP production [68].

Exploring the dynamics of cellular ATP generation in response to PBM therapy might inform the optimization of treatment protocols. For instance, studies on human neuronal cells (808 nm, 0.05 J/cm^2^) [69] and mouse muscle cells (630 + 850 nm, 2.5 J/cm^2^) [70] unveiled distinct temporal dynamics, with maximal ATP production observed at 10 min and 3–6 h post-irradiation, respectively. While these in vitro experiments highlighted the transient bio-stimulatory effects of PBM, recent endeavors led by Mintzopoulos et al. [71] assessed cortical phosphocreatine (PCr) levels and PCr/β-nucleoside triphosphate (β-NTP) ratios in dogs utilizing phosphorus magnetic resonance spectroscopy (31P-MRS) after acute and chronic transcranial laser therapy (808 nm). Intriguingly, immediate post-irradiation analyses unveiled no significant alterations in PCr levels or PCr/β-NTP ratios after a single session. However, repeating the irradiation sessions over two weeks demonstrated prolonged beneficial effects, suggesting enhanced neuronal bioenergetics.

### 6.2. Cerebral Blood Flow (CBF) and Angiogenesis

Diminished blood flow within the cerebral vasculature often represents one of the initial indicators of various brain disorders [72,73,74,75]. Nitric oxide (NO), a potent vasodilator, has the potential to be released from its binding sites during PBM via photodissociation in the respiratory chain. According to preclinical studies, PBM can widen blood vessel diameter, elevate NO levels in the brain, and enhance CBF [76,77,78]. Hence, it can be inferred that targeted PBM therapy on specific brain regions may influence regional CBF [8].

Experiments targeting mouse brain mitochondria revealed that LED light at 590 nm boosted NO production via CCO/NO activity modulation, while light at 627 nm and 660 nm had no significant effect [79]. Moreover, Uozumi et al. [77] proposed that following brain PBM therapy, the transient rise in CBF relies on nitric oxide synthase (NOS) activity and NO concentration. They demonstrated that over a 45 min irradiation period, transcranial NIR PBM at 808 nm elevated CBF by 30% in the illuminated hemisphere and 19% in the opposite hemisphere, accompanied by a 50% rise in cortical NO concentration. Additionally, pre-exposure to red LED light at 610 nm increased CBF 30 min post-reperfusion in a mouse model of cerebral ischemia [78].

Furthermore, several clinical investigations have elucidated how transcranial LED therapy affects CBF. In a study, Nawashiro et al. [80] documented that bilateral LED treatment (850 nm) applied to the forehead increased CBF in a vegetative patient’s left anterior frontal lobe by 20%. Similarly, Salgado et al. [81] revealed significant enhancements in the blood flow velocity following transcranial LED irradiation (627 nm) in the left middle cerebral artery (by 30%) and the basilar artery (by 25%) of healthy individuals. Additionally, Schiffer et al. [82] reported a notable increase in prefrontal CBF immediately after a single session of LED treatment to the forehead (810 nm) in patients with major depression and anxiety, though this effect was not sustained for two to four weeks after irradiation. Discrepancies in these findings may stem from differences in therapeutic and optical parameters, including treatment session frequency, wavelengths, and irradiation area.

So far, a multitude of animal and human research endeavors have demonstrated improvements in cerebral energy production and enhanced oxygen consumption consequent to transcranial PBM. Notably, transcranial PBM utilizing red and NIR wavelengths results in an increased consumption of cerebral O_2_, observed in both naïve rats [57] and AβPP transgenic mice [62]. Moreover, recent clinical investigations revealed that during and after transcranial laser therapy (1064 nm), a significant enhancement was observed in the oxygenation and hemodynamics of the brain [83].

PBM has emerged as a potent modulator of angiogenesis, thereby facilitating improved blood flow.

A study by Cury et al. [84] elucidated that the application of PBM at 780 nm and 40 J/cm^2^ stimulated an upregulation in the expression of hypoxia-inducible factor 1α (HIF1α) and vascular endothelial growth factor (VEGF), orchestrating the formation of new blood vessels in response to physiological demands. Furthermore, PBM demonstrated a remarkable ability to reduce the activity of matrix metalloproteinase 2 (MMP-2), thereby effectively promoting a microenvironment conducive to angiogenesis.

### 6.3. Oxidative Stress

Mitochondria are significant contributors to oxidative stress as they generate reactive oxygen species (ROS) that detrimentally impact neurons by impairing their mitochondrial functionality. A burgeoning body of literature has revealed links between oxidative stress and neurological conditions like AD, TBI, stroke, and major depressive disorder (MDD) [85,86,87,88]. The beneficial or adverse impact of PBM is intricately tied to mitochondrial ROS production. While low PBM doses induce mild mitochondrial ROS, affecting cellular signaling [89,90], higher doses, like 120 J/cm^2^, can prompt excessive ROS generation, potentially activating cellular apoptosis [91].

The prolonged and excessive release of NO can exert neurotoxic effects, potentially leading to tissue damage. Conversely, intriguing findings suggest that red light irradiation (660 nm) in rats following ischemic events can suppress nitric oxide synthase (NOS) activity [92], impacting various NOS isoforms including endothelial, neuronal, and inducible NOS. This suppression, along with the enhancement of total antioxidant ability by 808 nm laser irradiation [60], is proposed as a regulatory mechanism of oxidative stress by PBM. Additionally, blue laser irradiation at 405 nm unexpectedly increased superoxide dismutase as well as catalase on the HT7 acupuncture point while acetylcholinesterase action in the rat hippocampus was decreased [93]. Given mitochondria’s central role in red/NIR light–cell interactions, brain PBM emerges as a crucial initial intervention for ameliorating mitochondrial dysfunction induced by oxidative stress.

### 6.4. Neuroinflammatory Suppression

Neuroinflammation, a hallmark of brain disorders, is primarily facilitated by activated microglial cells. Following various neuronal injuries, microglia undergo a cascade of morphological and proliferative changes, culminating in the secretion of pro-inflammatory mediators such as cytokines, chemokines, NO, and ROS [94,95]. The excessive production of ROS triggers transcription factor NF-κB to translocate to the nucleus, which in turn stimulates the upregulation of pro-inflammatory cytokines [96]. PBM intervenes in this inflammatory cascade by impeding NF-κB signaling pathways, thereby lowering the release of pro-inflammatory cytokines and mitigating inflammatory reactions [97,98].

PBM has been shown to reduce pro-inflammatory cytokines such as IL-1β, IL-6, TGFβ, and TNFα [60,99,100]. These cytokines may be downregulated through the dephosphorylation of p38, as observed with PBM [101]. Moreover, in primary cortical astrocytes of rats, PBM has been found to inhibit Aβ-induced reactive oxygen species (ROS) originating from NADPH oxidase (NOX), as reported by Yang et al. [99]. ROS generated through NOX could activate various molecules as well as signaling pathways, including the p38 MAPK/Phospholipase A2 (cPLA2) pathway. Notably, investigations into astrocytes confirm that PBM intervenes in this cascade by mitigating cPLA2 phosphorylation, thus contributing to inflammation [102]. Furthermore, PBM exhibits a profound ability to downregulate the expression of ROS-generated inducible nitric oxide synthase (iNOS), further underscoring its anti-inflammatory potential [99,100].

In addition to the aforementioned findings, surprisingly, an excessive regimen of laser treatment sessions (administered daily for two weeks) elevated the expression of glial fibrillary acidic protein (GFAP) in mice with TBI. This surge in GFAP expression transiently hindered the repair mechanism in the brain subventricular zone (SVZ), although the beneficial impacts of PBM persisted in the long term [103]. This finding led to the notion that brain PBM’s anti-inflammatory properties could partly stem from its capacity to regulate microglial activity, subsequently leading to a reduction in inflammatory mediators.

### 6.5. Anti-Apoptosis and Neuroprotection

Apoptosis, a crucial process in normal brain aging and neurodegenerative diseases, operates through various pathways, with the intrinsic or mitochondrial pathway being significant in programmed cell death. A reduction in MMP triggers this process, which releases cytochrome c from mitochondria entering the cytoplasm and ultimately activating caspase-3, a key player in apoptosis [104]. The regulation of apoptosis involves pro-apoptotic (e.g., Bax) and anti-apoptotic (e.g., Bcl-2) proteins, where an increased expression of Bax or an elevated Bax/Bcl-2 ratio triggers the caspase cascade, prompting the apoptosis process [105]. There is abundant evidence indicating that PBM can be employed for neuroprotection, enhancing the survival and lifespan of neuronal cells by significantly ameliorating neuronal apoptosis through anti-apoptotic effects.

Research findings indicate that LED therapy (670 nm) administered twice daily yields a significant reduction in apoptotic events among both striatal and cortical neurons subjected to rotenone and MPP+ exposure [106]. Prior exposure to 670 nm light, at fluences 4 J/cm^2^ [107] and 30 J/cm^2^ [108], effectively protects primary neurons from apoptosis triggered by various neurotoxins. Moreover, the neuroprotective impacts of PBM, particularly near the 810 nm wavelength, have been demonstrated in vitro, ameliorating mitochondrial structural damage and preventing MMP collapse in neurotoxicity models [109]. Both red light (670 nm LED) [110] and NIR light (810 nm laser) [109] have demonstrated substantial reductions in neuronal apoptosis by inhibiting caspase-3 activity and mitigating the activity of pro-apoptotic factors, including Bax and BAD. This efficacy is postulated to stem from the initial light absorption mechanism of mitochondrial inner membrane enzyme chromophores, which enhances MMP stability and hence confers neuroprotection [111].

Moreover, the regulation of apoptosis is significantly influenced by the serine/threonine kinases that comprise the protein kinase C (PKC) family. Activation of PKC can modulate the expression of Bax and Bcl-xl in cells, leading to the suppression of apoptosis [112,113]. By activating the PKC pathway and reducing the Bax/Bcl-xl mRNA ratio, Zhang et al. [114] demonstrated that low-dose laser radiation at 632.8 nm may successfully reverse PC12 cell apoptosis.

In addition to the aforementioned findings, PBM is thought to engage alternative anti-apoptotic mechanisms. Laser irradiation at 632.8 nm effectively suppressed the activities of caspase-3, glycogen synthase kinase (GSK-3β), and Bax, thus preventing staurosporine-induced cell apoptosis by deactivating the GSK-3β/Bax pathway [115]. Furthermore, PBM therapy at 632.8 nm has been proposed to counter PC12 cell apoptosis by activating the Akt/YAP/p73 [116] and/or Akt/GSK3β/β-catenin pathways [117].

### 6.6. Neurogenesis and Synaptogenesis

The potential of PBM to enhance both synaptogenesis and neurogenesis is one of its most remarkable and substantial effects on the brain. PBM facilitates neuronal connectivity by intricately regulating the expression and function of different neurotrophic factors (neurotrophins) including brain-derived neurotrophic factor (BDNF), glial cell-derived neurotrophic factor (GDNF), and nerve growth factor (NGF). PBM emerged as a potential approach to mitigate cortical dendritic atrophy in AD progression by augmenting BDNF expression [118]. Stimulating the ERK/CREB/BDNF pathway following PBM at 632.8 nm has been suggested to rescue dendritic atrophy [118]. Furthermore, PBM with the same laser wavelength triggered the activation of intracellular IP3 receptors, leading to increased intracellular Ca^2+^ levels with subsequent ERK/CREB pathway activation, ultimately enhancing BDNF expression [119]. Meanwhile, in vivo studies have demonstrated that coherent laser light at 670 nm significantly improves the expression of BDNF in rats’ occipital cortex [110]. Similarly, LED therapy at 660 nm caused the inhibition of apoptosis triggered by oxidative stress in hippocampal cell lines, coupled with a notable increase in BDNF expression [120]. Additionally, PBM therapy applied intracranially with non-coherent LED light at 670 nm in a primate model of PD led to an increased expression of GDNF in the striatum, correlating with behavioral improvements [121].

Oron et al. provided the first evidence of PBM-stimulated neurogenesis and neuronal progenitor cell migration in rats in vivo [122]. Their study demonstrated that PBM at 808 nm significantly increased proliferating cell numbers (through the incorporation of BrdU) in the SVZ of the hemisphere affected by stroke. Additionally, in a rat ischemic model, laser irradiation at 650 nm on acupuncture points GV20 and HT7 ameliorated cognitive impairment by upregulating CREB along with BDNF gene expression in the hippocampus [123]. In further investigations on TBI mice, Xuan et al. clarified the optimum transcranial PBM regimen at 810 nm for neuroprotection. They discovered that one or three days of laser irradiation triggered neurogenesis, enhanced synaptogenesis and neuroplasticity markers (synapsin-1) within the cerebral cortex, and boosted BDNF in both DG and SVZ [124,125]. Moreover, PBM has the capacity to stimulate dormant neural stem cells located in specific areas of mature organisms’ brains. The activation of these stem cells could promote tissue regeneration. PBM can also stimulate the generation of neuroprogenitor cells, similar to neural stem cells, which could positively impact neurogenesis [125].

### 6.7. Impacts on Intrinsic Brain Networks

Intrinsic brain networks are a complex of distant yet interconnected networks that exist within the brain, among which the default mode network (DMN), salience network (SN), and central executive network (CEN) are most prominent. These networks are active not only when stimulated by external inputs but also during rest, suggesting a coordinated regulation of brain activity [126]. They play significant roles in modulating higher cognitive and emotional functions, but disruptions in their activity, seen in conditions like chronic neurodegenerative diseases or acute brain injuries, lead to imbalances [127].

For instance, patients with traumatic brain injury (TBI) often show weakened links within and between DMN, SN, and CEN nodes, leading to cognitive impairment [128,129]. It is hypothesized that targeted light therapy, matching specific brain regions, could potentially restore these functions, offering enhanced therapeutic benefits [130]. Studies conducted by Naeser et al. demonstrate promising outcomes from transcranial LED therapy for enhancing the cognitive functions of TBI patients, targeting DMN, SN, and CEN nodes, and likely by boosting metabolic capacity within these networks [131]. On top of that, they theorized that PBM therapy may alleviate post-traumatic stress disorder (PTSD) symptoms by modulating DMN and SN activities [130]. Additionally, PBM therapy has shown neurotherapeutic efficacy in stroke patients with aphasia by stimulating cortical nodes within the CEN network [132].

## 7. Systemic Effects of PBM Therapy

While direct brain PBM is considered the predominant therapeutic approach, there is a burgeoning understanding of neuroprotective effects stemming from irradiating specific non-brain regions. This implies that the light stimulation of certain organs may yield systemic benefits that indirectly influence the brain [133]. Clinical investigations have unveiled intriguing outcomes: LED irradiation at 660 and 850 nm targeting the back and thighs relieved depression symptoms in individuals suffering from low back pain [134], while irradiation with laser (514 and 632.8 nm) on auricular acupoints and the neck ameliorated depression and mitigated alcohol withdrawal symptoms in those with alcohol addiction [135]. Additionally, the application of a near-infrared laser array stimulator (LAS) at 10 Hz on the palm revealed an increased power of alpha rhythms and theta waves primarily in the occipital, parietal, and temporal regions of the brain, persisting for at least 15 min [136]. Such effects hold potential for clinical applications, such as insomnia therapy. Furthermore, in a murine model of PD, administering PBM to remote tissues, excluding the head, showed a remarkable neuroprotective effect by preserving dopaminergic neurons in the midbrain [137].

The precise mechanisms underlying the systemic response to PBM therapy remain elusive, but potential factors include enhanced immune cell activity [138], the modulation of cytokines [139], and increased mitochondrial ATP levels in platelets. Additionally, mesenchymal stem cell migration, which is induced at locations of brain damage, may have a neuroprotective abscopal impact [140,141]. Mesenchymal stem cells may be mobilized via PBM of the bone marrow, allowing them to migrate to the brain and potentially restore cognitive function in AD patients as the disease progresses [142,143]. Light absorbed by other tissues prior to reaching the brain, akin to remote action, may contribute to neuroprotective effects.

In addition to remote radiation therapy, investigations on animals with laser or LED devices have demonstrated the neuroprotective effects of full-body PBM therapy [110]. For instance, direct LED light application (710 nm) on the top surface of an animal’s cage reduced microglial activation, stimulated cellular immunity, and lessened the size of the brain infarction, leading to neurological improvements in a rat stroke model [144]. Long-term exposure to white fluorescent light in the mouse SNc reduced dopaminergic neurons, but this was not the case with LED PBM treatment at 710 nm [145]. Furthermore, full-body PBM therapy with LED (1072 nm) improved working memory in middle-aged mice [146] and reduced Aβ plaque deposition in transgenic AD mice [147]. Particularly, noninvasive multisensory 40 Hz stimulation, comprising combined light and sound stimulation, reduces amyloid deposition across the cortex, notably the prefrontal cortex, thereby enhancing neural activity in an Alzheimer’s disease mouse model [148].

## 8. Clinical Applications of Brain PBM Therapy

From the medical perspective, a wide array of disorders afflicts the brain, and are broadly categorized into four groups: traumatic events, degenerative diseases, and neurodevelopmental and psychiatric disorders that could potentially benefit from various forms of transcranial photobiomodulation therapy (Figure 4).

Clinical brain PBM therapy trials conducted recently have addressed TBI, ischemic stroke, AD, PD, and MDD. However, there is also increasing interest in exploring this non-invasive technique to enhance cognitive function in completely healthy people (Table 1).

### 8.1. Stroke

The initial major investigations into PBM for stroke were conducted through the NeuroThera Effectiveness and Safety Trials (NESTs), which comprised three clinical trials involving 1410 participants with 1 J/cm^2^ and 808 nm radiation applied to their brains for 16 to 18 h following an ischemic stroke. While NEST-3 was halted due to its anticipated insignificance, NEST-1 and NEST-2 showed promising trends. Of those who received genuine treatment in NEST-1, 70% exhibited successful results compared with 51% in the sham group. Similarly, in NEST-2, a greater proportion of patients who received genuine therapy reported a positive outcome compared with those who had the sham treatment, although this lacked statistical significance [160]. However, the termination of NEST-3 might be attributed to factors like inadequate light penetration into the brain or from the choice to employ a single tPBM treatment rather than several treatments [161]. Despite this, all three studies demonstrated the safety of PBM with minimal adverse effects.

Additionally, some studies in patients with chronic stroke have aimed to demonstrate the neuroprotective or neuroreparative potential of PBM therapy using methods such as transcranial [132] and multi-area [150] irradiation.

### 8.2. Acute and Chronic TBI

While most animal studies predominantly focus on acute models of TBI, clinical research primarily centers on chronic TBI patients. Individuals recovering from moderate-to-severe head injuries commonly endure a myriad of persistent symptoms like cognitive dysfunction (such as memory deficits, impaired executive function, and attention difficulties), persistent headaches, sleep disturbances, and depressive symptoms.

Recent ongoing research exploring TBI has revealed that transcranial LED treatment (633/870 nm) can enhance sleep quality, social functioning, and self-awareness [131]. Moreover, employing NIR laser therapy at high fluence has exhibited enhanced clinical efficacy, manifesting in reduced headache symptoms, improved sleep quality, and enhanced cognitive and emotional well-being among TBI patients [26]. Furthermore, transcranial PBM therapy at an unconventional wavelength of 785 nm has demonstrated notable success in ameliorating alertness and consciousness in TBI patients with severe cognitive impairments [151].

### 8.3. Alzheimer’s Disease

Although numerous animal studies have been conducted, research on the success of PBM therapy remains scarce in patients with AD and dementia. However, the limited human studies available have shown that NIR PBM therapy led to notable changes in various aspects of patient well-being, including mood states, sleep quality, EEG patterns, and cognitive performance, like memory and attention [35,162]. Additionally, the application of red laser therapy in AD patients through an artery catheter directed towards the brain improved CBF while significantly decreasing dementia ratings [152].

### 8.4. Parkinson’s Disease

In recent years, transcranial PBM therapy has shown promise in treating illnesses like TBI, stroke, and depression. Most of the clinical studies that have examined this therapy have focused on the cortical parts of the brain. However, anomalies in the SNc, a midbrain region situated beneath the dura and 80–100 mm below the coronal suture, are associated with the etiology of PD. Research has indicated that NIR light might not effectively enter the human brain beyond 20 mm from the cortical surface, presenting a notable challenge for transcranial PBM therapy in PD treatment [32]. Nevertheless, despite this significant obstacle, a solitary non-controlled, and non-randomized investigation highlighted enhanced motor and cognitive abilities following a two-week course of tPBM therapy in PD patients [153].

### 8.5. Depression and Related Psychiatric Disorders

Efforts towards developing efficient and sustainable treatments for major depression have been ongoing globally for many years. Current investigations on the antidepressant impacts of PBM therapy have so far only had short follow-up intervals and can fall into two main categories: those focusing on MDD patients [82,154,155] and those focusing on TBI patients with comorbid depression [26,130,163].

In the initial study involving MDD patients, LED therapy in a single session showed promise in reducing the symptoms of anxiety and depression (measured using Hamilton scales) two weeks after irradiation [82]. Similarly, transcranial LED therapy led to substantial reductions in PTSD scores and depression symptoms after just one week of treatment. However, the treatment response over two months did not show a consistent trend [131].

In addition to LED light therapy, transcranial high-power lasers have demonstrated efficacy in alleviating depressive symptoms in patients with MDD [155] and those with TBI who also had concomitant depression [26] 6–8 weeks after radiation therapy. Furthermore, a study by Disner et al. [131] indicated that transcranial laser treatment directed to the right forehead was superior to PBM applied to the left forehead in terms of reducing depressive symptoms. This finding may suggest that PBM has a different effect on different brain areas in MDD patients.

### 8.6. Cognitive Improvement among Healthy Subjects

In the realm of daily mental engagements, the intricate web of cognitive processes involved the domains of short-term and long-term memory retention, the complexities of decision-making, the sustained vigilance of attention, problem-solving, and strategic planning, all underpinned by the orchestration of executive functions [164,165]. As people age, there is often a decline in cognitive functions, making the enhancement or preservation of these abilities particularly important, especially for older adults who face increased susceptibility to dementia [166]. In this context, the burgeoning interest in the potential cognitive augmentation by PBM therapy emerges as a beacon of hope in the years to come. Noteworthy studies by Gonzalez-Lima and his research cohorts, employing a 1064 nm laser, have revealed a spectrum of cognitive enhancements in healthy young cohorts, spanning from the augmentation of prefrontal rule-based learning to bolstering sustained attention, short-term memory, and the intricate mechanics of executive functions [156,167,168]. Moreover, according to their research, PBM therapy targeted towards the prefrontal region might enhance the brain’s hemodynamics as well as oxygenation, which are critical to higher-level cognitive functions [159].

## 9. Conclusions and Future Prospects

The future prospects of PBM in the field of neuroscience appear exceedingly promising, owing to its remarkably positive outcomes in animal models and the encouraging results from initial clinical trials. Given the considerable challenges encountered in developing pharmaceutical interventions for conditions like stroke [169] and TBI [170], where progress has been largely elusive despite substantial investments, there is a compelling case for a broader exploration of PBM’s potential. Additionally, the stagnation in discovering effective therapies for neurodegenerative disorders like Alzheimer’s and Parkinson’s diseases underscores the urgency of considering alternative approaches like PBM [171]. Furthermore, the favorable outcomes demonstrated by brain PBM therapy in mitigating symptoms of depression and anxiety may pave the way for novel trials targeting a broader range of psychiatric disorders including bipolar disorder, attention deficit hyperactivity disorder (ADHD), obsessive compulsive disorder (OCD), as well as schizophrenia and autism.

Considering PBM’s well-documented safety profile and its notable absence of adverse effects, alongside its relative cost-effectiveness, there is a strong argument for initiating comprehensive clinical trials for these indications. Furthermore, the disappointingly low efficacy rates and often distressing side effects associated with conventional psychiatric medications, such as antidepressants [172], highlight the need for novel therapeutic modalities like PBM. While further research is indispensable, the accessibility of affordable laser or LED devices in the clinic or home for PBM therapy suggests it can be anticipated to emerge as a pre-eminent approach for neurorehabilitation in the years ahead.

## Figures and Tables

**Figure 1 cells-13-00966-f001:**
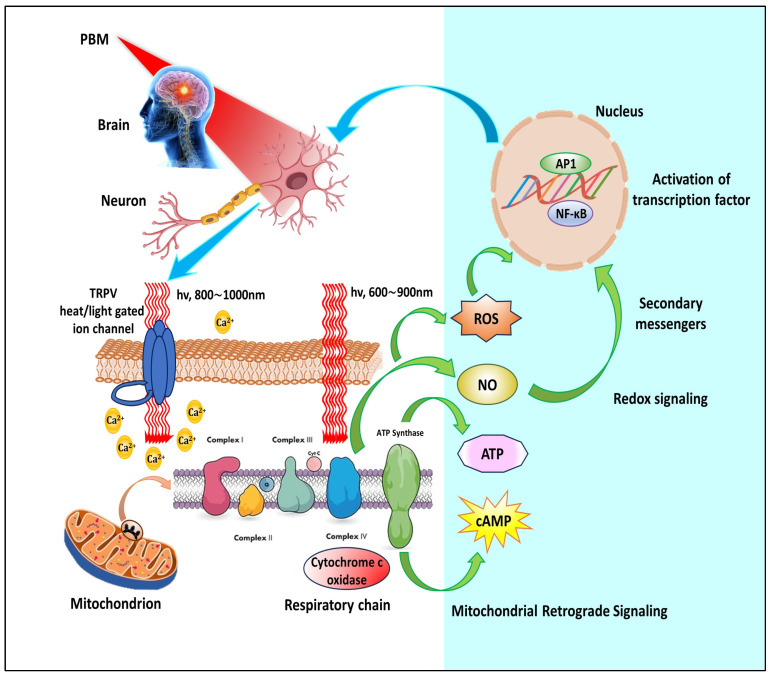
Cellular and molecular mechanisms underlying photobiomodulation. AP1, activator protein 1; ATP, adenosine triphosphate; Ca^2+^, calcium ions; cAMP, cyclic adenosine monophosphate; NF-κB, nuclear factor kappa B; NO, nitric oxide; ROS, reactive oxygen species; TRPV, transient receptor potential vanilloid.

**Figure 2 cells-13-00966-f002:**
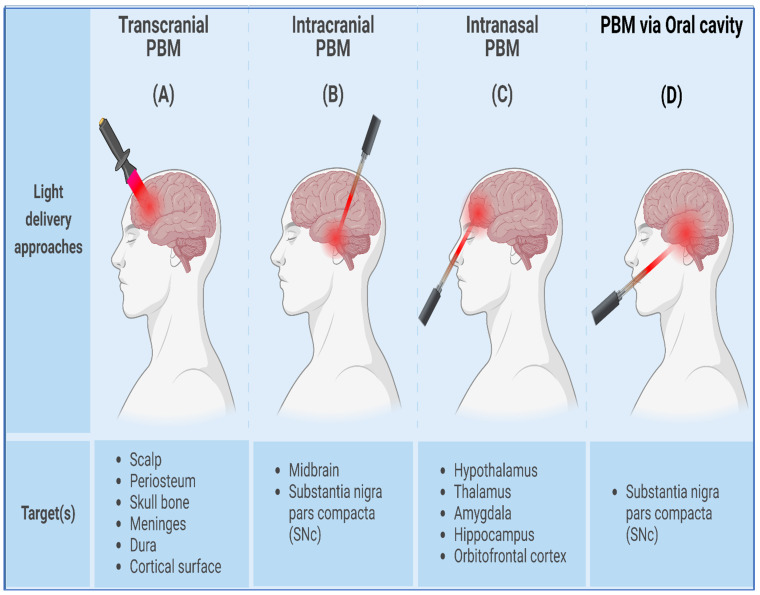
Different light delivery approaches for effective brain PBM therapy. (**A**) Transcranial, (**B**) intracranial, and (**C**) intranasal PBM therapy; (**D**) brain PBM therapy through oral cavity.

**Figure 3 cells-13-00966-f003:**
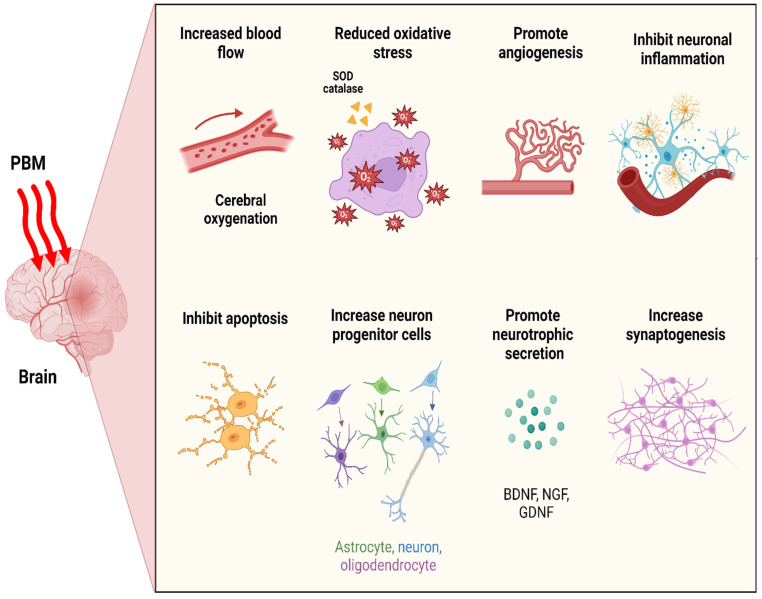
Functional processes specific to brain tissue following photobiomodulation therapy. BDNF, brain-derived neurotrophic factor; GDNF, glial-derived neurotrophic factor; NGF, nerve growth factor; SOD, superoxide dismutase.

**Figure 4 cells-13-00966-f004:**
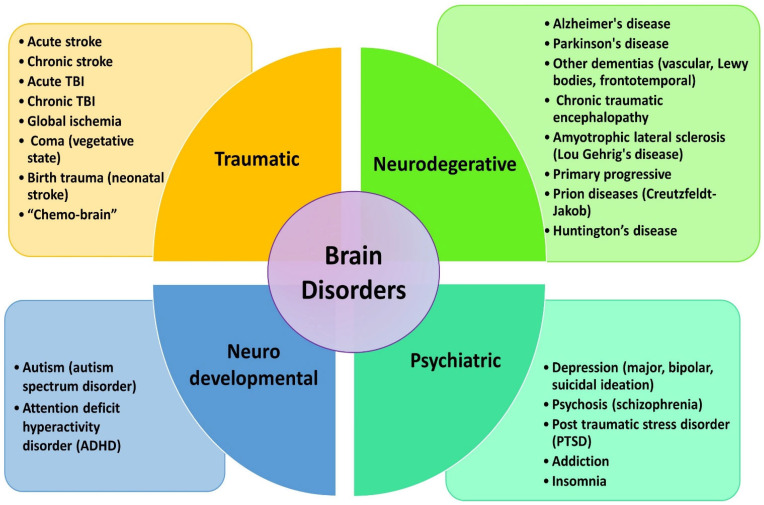
Types of brain disorders susceptible to potential treatment via transcranial photobiomodulation (tPBM).

**Table 1 cells-13-00966-t001:** Reports of clinical studies demonstrating neuroprotective impacts of PBM in brain disorders.

Subjects (n)	Parameters	Major Findings	References
Acute stroke (120)	808 nm; 10 mW/cm^2^, 1.2 J/cm^2^ laser applied at cortex, 2 min for each site, CW	Induced positive neuroprotective effects within 24 h of irradiation after stroke onset.	[149]
Chronic stroke (1)	660 and 850 nm; 1400 mW, 2.95 J/cm^2^ LEDs delivered to multiple areas (32 sites), 1 min for each site, spot size of 0.196 cm^2^, 1/week for 8 weeks	Significant changes across all areas of deficits and notable enhancements in clinical manifestations of physical symptoms.	[150]
Chronic TBI (2 with depression) (11)	633 and 870 nm; 500 mW, 22.48 mW/cm^2^, 13 J/cm^2^ LEDs administered at 11 sites of midline and bilateral forehead, 10 min per site, 3×/week for 6 weeks, CW	Enhanced sleep patterns, diminished symptoms of post-traumatic stress disorder, and heightened proficiency across social, interpersonal, and occupational domains.	[131]
Chronic TBI (6 with MDD) (10)	810 and 980 nm; 10 and 15 W, 14.8–28.3 J/cm^2^ laser applied in unilateral (forehead) and bilateral areas (prefrontal and temporal), 8–12 min per site, 2–3×/week for 8 weeks, with PW at 10 Hz	Reduced headache, sleep disruption, mood instability, anxiety levels, irritability, and cognitive symptoms.	[26]
TBI with disorders of consciousness (5)	785 nm; 10 mW/cm^2^ laser delivered on the superior crest level of the fossa sphenoidale on the forehead, 10 min, 5×/week for 6 weeks, CW	Increased consciousness and alertness; as an adverse effect, epileptic episodes occur.	[151]
Alzheimer’s disease (89)	Visible region of spectrum; 20 mw laser conjugated with an optic fiber with a diameter of 25–100 μm inserted through a catheter into the femoral artery, navigated to reach the distal sites of the anterior and middle cerebral arteries, 20–40 min, CW or PW, or combined modes.	Reduced irreversible dementia; enhanced microcirculation within the cerebral region, leading to cognitive rehabilitation.	[152]
Parkinson’s disease (8)	Laser applied 2 weeks daily in several areas (bilateral occipital, parietal, temporal, frontal lobes and along sagittal sutures)	Enhanced stability, walking pattern, mitigation of freezing episodes, cognitive capabilities, movement while in bed, and speech challenges.	[153]
Major depressive disorder (10)	810 nm; 250 mW/cm^2^, 60 J/cm^2^ LEDs applied at right and left forehead, 4 min, one irradiation session, CW	Reduced rates of depression and anxiety for two weeks after irradiation; no significant impacts on cerebral blood flow.	[82]
Major depressive disorder (4)	808 nm; 5 W, 700 mW/cm^2^, 84 J/cm^2^ laser administered at right and left forehead center at 20 and 40 mm from the sagittal line, 2 min per site, 2×/week for 3 weeks, CW	Occurrence of depression exhibited a notable decrease around six to seven weeks following the irradiation.	[154]
Patients with elevated depression symptoms (51)	1064 nm; 250 mW/cm^2^, 60 J/cm^2^ laser delivered to the medial and lateral parts of the left or right side of the forehead, 4 min per site, for 2 sessions, CW	Prefrontal irradiation on the right side enhanced the efficacy of attention bias modification interventions and alleviated symptoms of depression.	[155]
Healthy volunteers (40)	1064 nm; 250 mW/cm^2^, 60 J/cm^2^ laser applied at right frontal pole 4 cm medially and laterally, 4 min, one irradiation session, CW	Enhancements in reaction time during the Psychomotor Vigilance Task and performance in a delayed match-to-sample memory task; sustained positive emotional states evident even two weeks after the treatment.	[156]
Healthy volunteers (14)	905 nm; 50 mW/cm^2^ laser applied over the primary motor cortex (M1) areas, 3 J/cm^2^ per site, 60 s, PW at 3000 Hz	Temporary decrease in motor cortex excitability.	[157]
Healthy volunteers (55)	810 nm, 500 mW/cm^2^ at scalp via 4 laser needles applied over the primary motor cortex (M1) area, 10 min, one irradiation session, CW	Reduced magnitude of motor-evoked potentials, heightened short-interval cortical inhibition, and diminished facilitation.	[158]
Healthy volunteers (11)	1064 nm; 250 mW/cm^2^ laser applied at right forehead, 13.75 J/cm^2^ per 1 min for 8 min, one irradiation session, CW	During and after irradiation, cerebral levels of oxidized CCO, oxygenated hemoglobin, and total hemoglobin were elevated.	[159]

CCO, cytochrome c oxidase; CW, continuous wave; LEDs, light-emitting diodes; MDD, major depressive disorder; PW, pulsed wave; TBI, traumatic brain injury.

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
