# Peer review of "Photobiomodulation Therapy on Brain: Pioneering an Innovative Approach to Revolutionize Cognitive Dynamics"

_cells, 2024, doi:10.3390/cells13110966_

Round 1

Reviewer 1 Report

Comments and Suggestions for Authors

Thank you for this opportunity to review this article. My understanding of this review article is that it explores the effects of photobiomodulation (PBM) on the brain. It reviews the use of PBM as a therapeutic modality to delay disease disease progression. 

The simplified mechanism involves activation of cellular chromophores via near infrared light to induce redox cascades within cells to accelerate neuronal activity. The research on the field started with an accidental discovery my McGuff and had applications in other organ systems, but in the past 20 years it has proven to enhance cognitive function. It helps in acceleration of neurogenesis using light spanning wavelength from the red spectrum to the near-infrared (NIR) region, between 600 to 1100 nm. Currently, there seems to an abundance of supportive animal studies but clinical studies for transcranial PBM are lacking. 

The article describes the molecular mechanism - the cascade of light activation promotes ATP production. The main targets are cytochrome C oxidase and nitric oxide. However, there might be additional areas or mechanisms in the cellular matrix that might be affected by light therapy, whether in the infrared spectrum or green spectrum. 

The methods of delivery have been described as transcranial PBM - limitations include decrease in light intensity as it traverses various fascias. Other therapies mentioned to address this limitation include TRUE encoding and nanoparticles. An intracranial approach describes a more targeted delivery, to allow for deeper tissue penetration that contributes to treatment modalities for parkinson’s. Another approaches are intranasal and intraoral; via implantation of a fiberoptic source. 

Figure 3 very well describes the functional processes. 

It has been described from a bioenergetics perspective via reference to animal studies. 

The paper also describes various mechanisms of neuroinflammatory suppression. It talks about neuroprotective mechanisms via LED therapy and synaptogenesis in rat models for improvement in stroke. 

Section 5.7 aptly desribes the effects of targeted light therapy on patients with traumatic brain injury.

Influence on other systems is described in section 6; it is yet to be completely understood but hypothesis is towards improved immune function and stem cell stimulation that can have a positive effect on neural activity. 

As an obvious progression, the article describes the clinical implications. I found table 1 to be most informative. 

My only comment to the very well structured article would be a description of method section. Even though this is a scoping review, it would be beneficial to know how the review was conducted and how many articles were reviewed and whether there was a criteria for selection. I think it would add to the validity of the study. 

Author Response

Dear Reviewer

Thank you for your valuable comments.

Reviewer 2 Report

Comments and Suggestions for Authors

The manuscript by Nairuz et al describes in a very exhaustive manner the effect of Photobiomodulation Therapy on Brain as Pioneering an Innovative Approach to Revolutionize Cognitive Dynamics.

In my opinion the manuscript is well conceived and written since it includes the description of all the aspects relate to Photobiomodulation therapy including: the delivery, the mechanisms of action of the neurobiological and clinical effects, the potential therapeutic uses and the safety profile. The figures and the tables are usefully and exhaustively drive the reader in the comprehension of the aim of the study.

Author Response

Dear Reviewer,

Thank you so much for your kind and motivational words. We extend our heartfelt gratitude for your time and efforts. Your comments and nice words encourage the authors for future work.

Reviewer 3 Report

Comments and Suggestions for Authors

The authors of this paper focus on reviewing the articles treated brain problems with photobiomodulation. I believe the strength of this article can be improved by addressing the following concerns:

Comments:

1.     The abbreviation of some words should exist at the first time. For example, line 66-67, it should be modified as “spanning from stroke and traumatic brain injury (TBI) to Parkinson’s disease(PD), Alzheimer’s disease(AD), and depression,..”, same as line 318, MDD(major depressive disorder).

“PTSD” in line 561 should list the full name, please add the full name of the “PTSD”.

2.     The Figure 1 has some mistakes, for example, the arrow from ATP synthase to NO is wrong, because NO is released from CCO, however NO is induced by iNOS, as you mentioned in line 38-41. Please redraw Figure 1.

3.     The values in line 177: (23:6+0:7 mm) [44] are not clearly indicated.

4.     Reference 47 indicates that only 4.2% of the light penetrates through the human skull. According to Reference 25, at wavelengths of 810 nm and 980 nm, only 0.45% and 2.9%, respectively, penetrate 3 cm of ex vivo skull. Although Table 1 summarizes the effects reported in previous studies, the power levels are indeed very low. If this method is to be viable for future applications, it is crucial to address whether the accumulated dose over the irradiation time is an important parameter. Additionally, are there any studies in the literature that support this?

5.     Line 482-485, should be deleted (including reference 149-150), because the name of the device or the company is not suitable to exit in this article. The device doesn’t have evidence to directly affect the brain activity.

6.     There is a published paper about the stimulation of brain, the specific brainwave can be induced with an array NIR lasers [1]. This paper can be cited in 6.Systemic effects of PBM therapy. This is a new method to affect brain activity by radiating non-brain regions with PBM therapy. This article should be cited for completeness.

1.       Wu, J.H.; Chang, W.D.; Hsieh C.W. et al. Effect of low level laser stimulation on EEG. Evidence-Based Complementary and Alternative Medicine 2012, vol. 2012, Article ID 951272, 11 pages.

Author Response

Dear Reviewer,

Thank you for your time and valuable comments.
